# Humanness Is Not Always Positive: Automatic Associations between Incivilities and Human Symbols

**DOI:** 10.3390/ijerph18084353

**Published:** 2021-04-20

**Authors:** Laura Rodríguez-Gómez, Naira Delgado, Verónica Betancor, Xing Jie Chen-Xia, Armando Rodríguez-Pérez

**Affiliations:** Department of Cognitive, Social and Organizational Psychology, Faculty of Psychology, University of La Laguna, 38200 San Cristóbal de La Laguna, Santa Cruz de Tenerife, Spain; lrodrigg@ull.edu.es (L.R.-G.); ndelgado@ull.edu.es (N.D.); xchenxia@ull.edu.es (X.J.C.-X.); arguez@ull.edu.es (A.R.-P.)

**Keywords:** civic behavior, uncivil behavior, dehumanization, automatic associations, SC-IAT

## Abstract

Uncivil behavior involves an attack on social norms related to the protection of public property and respect for community life. However, at the same time, the low-frequency and relatively low-intensity damage caused by most of these behaviors could lead to incivilities being considered a typically human action. The purpose of this set of studies is to examine the automatic associations that people establish between humanness and both civic and uncivil behaviors. Across three studies, uncivil behaviors were more strongly associated with human pictures than animal pictures (study 1) and with human-related words than animal-related words (study 2). We replicated study 2 with uncivil behaviors that do not prime graphically human beings (study 3). Overall, our results showed that uncivil behaviors and civic behaviors were clearly associated with human concepts. Our findings have direct implications for the conceptualization of humanness and its denial.

## 1. Introduction

It is hard to imagine how human society could work without social norms. People need norms to guide their actions, provide order and regularity in social relationships, and understand each other’s acts [1,2,3]. These social norms define socially correct behaviors and proscribe unacceptable ones for a given social unit [4,5]. The present study focuses on one important type of social norm, civility, and its link with the perception of humanness.

Civility is essential for sustaining an urban life characterized mainly by unpredictable and transitory relations between strangers [6,7]. Civic behaviors respect social norms and guarantee survival. Civic behavior could be defined as a type of ethical behavior that includes courtesy, manners, good citizenship, and concern for the welfare of the community members [8]. On the other hand, uncivil behavior involves an attack on social norms related to the protection of public property, respect for the other, or community life [9]. Unlike criminal acts, uncivil behaviors are not so dangerous as to merit the attention of the police or constitute a reason for systematic repression. However, they do have important negative effects and often pose a threat to those affected [9,10]. In fact, different studies have shown that city residents think that uncivil behaviors are the most significant factors of urban stress and the ones that most reduce the quality of life in a community [11,12]. It has also been shown that people who experience uncivil behaviors are more likely to express feelings of anxiety and depression and experience health problems [13,14]. In this sense, incivility is framed not only as a concern for environmental deterioration but also as a phenomenon with implications for public health. The study of uncivil behaviors could help us understand what characteristics of these behaviors generate psychosocial effects on wellbeing and health so that these results can be incorporated into political, economic, and legal considerations related to environmental stewardship.

Uncivil behaviors have been related to deviation from the norm [15] to social control [9,16], to negative emotions [17], and to moral implications [18]. However, what is the link between incivilities and humanness? Are uncivil behaviors perceived as typically human? Are they considered to represent a lack of humanity? The purpose of this set of studies is to examine the automatic associations that people establish between humanness and both civic and uncivil behaviors.

The links between uncivil behaviors and what is considered inherently human could be explored from multiple perspectives. From an evolutionary approach, civility has been considered functional for human beings. The evolutionary arguments indicate that what is relevant in social terms is the community’s wellbeing and not the individual. In this way, moral behaviors are rewarded as socially beneficial behaviors, which allow the wellbeing and survival of the community, and immoral behaviors are rejected and condemned as those, which are harmful and accelerate the extinction of society [19]. Even more relevant to the studies presented here is the established association between civility and humanity. Literature on dehumanization highlights the idea that uncivil behaviors are considered the expression of a lack of “uniquely human” traits and hence correspond to lower humanity (see [20] for a review). In this sense, Haslam’s integrative review on dehumanization [21] defines two distinct forms of humanness: human uniqueness and human nature. Uniquely human (UH) characteristics define the traits that separate humans from the rest of animals: civility, refinement, moral sensitivity, rationality, and maturity. The human nature (HN) dimension may also be understood in terms of features that are “essentially, typically, or fundamentally” human. In other words, attributes that are typically human—emotionality, interpersonal warmth, agency, and flexibility—may not necessarily be the same ones that discern humans from other animals. HN is further distinguished from UH in that UH characteristics are viewed as socially acquired, the consequence of the cultural and societal environment in which these characteristics are developed. In contrast, HN is primarily concerned with inherent, universal characteristics, reflecting the essence of human beings independent of culture [22,23]. Therefore, according to the Haslam model, uncivil behaviors are considered as a lack of “uniquely human” traits and hence correspond to lower humanity on UH. That is, uncivil behavior should be associated with animalization. Importantly, this proposal is theoretical, and there is no empirical evidence to date that corroborates this rationale.

The literature on dehumanization also presents another opposing possibility: both types of behaviors (civic and uncivil) are closely associated with humanity. From the mind perception framework [24,25], two dimensions comprise what people consider to be a truly human mind: agency and experience. Agency includes mental capabilities, such as thinking, self-control, and communication, whereas experience comprises attributes, such as emotion, consciousness, and personality. Agency is primarily concerned with higher-order cognitive abilities, which can be understood as elements of unique humanness [20,26]. Closely related to this idea is the moral typecasting theory (MTT), which considers that the social world is understood in terms of moral agents and moral patients [25]. In this sense, the moral typecasting theory posits that, when a person commits a moral act, whether positive or negative, perceivers attribute to that person the qualities of a moral agent [25,27]. According to these theories, uncivil behaviors should be associated with humanity to the same extent as civility is.

However, while the relationship between uncivil behavior and humanity remains unexplored, research on human perception of perpetrators of harmful acts can be relevant. In these studies, the results tend to favor the dehumanization of the perpetrator compared to what is anticipated by moral typecasting theory. For example, Khamitov and colleagues [28] found evidence consistent with a dehumanization perspective on the way people conceptualize the agency of harmful agents. In fact, their findings were in line with previous research on dehumanization (e.g., [29]) that shows that harmful agents are perceived with less agency (and overall, fewer humanity traits) than both neutral agents and benevolent agents. These findings directly contrast with predictions from MTT (see also [27,30,31,32]). In the same line, Swiderska and Küster [33] contrasted predictions concerning attribution of mental dimensions from moral typecasting theory with the denial of agency from dehumanization literature. Across three experiments, they investigated mind attribution to robotic moral agents. The results consistently supported dehumanization theory over moral typecasting; specifically, malevolent robots were assessed to possess fewer mental capacities than benevolent and neutral robots. Interestingly, the harm inflicted by a comparatively lesser mind is perceived as less serious due to it not possessing the same moral weight and implications for the moral patient in the interaction. Could these results about harmful behaviors be extended to uncivil behaviors?

Other fields of study could suggest that uncivil behaviors are associated with human qualities and social advantages. For example, from a neurocognitive approach, there are surprising links between human capabilities and immoral behaviors. Persistent immoral behavior can be interpreted as an alternative evolutionary strategy that can be helpful at low rates in society [34]. While most studies have detected reduced activity in specific areas in the brain of immoral individuals, several studies highlight the idea that psychopaths have developed advanced cognitive processes for taking advantage of others in society [35] by having superior functioning in areas linked to adaptive strategies, such as lying [34]. Immoral behaviors are also involved in language, one of the most important characteristics differentiating humans from animals. Dor [36] defends the idea that lying made a significant contribution to the evolution of language, increasing its complexity and cognitive requirement. Moreover, uncivil behaviors such as dishonesty are also related to abilities as human as creativity. In this way, Gino and Wiltermuth [37] found that when people act dishonestly, they feel free from the rules, and they become more creative, which allows them to find more creative excuses for their immoral behavior. From a sociological approach, uncivil behavior has been treated as a demonstration of civil disobedience to the social norms imposed by life in society [38], focusing on the dysfunctionality of this type of behavior. However, an alternative perspective highlights the individual and collective functionality of uncivil behaviors. From this approach, uncivil behaviors sometimes constitute expressions of dissent. In other words, they represent an expression of opinion contrary to the existing socio-political hierarchy [39]. Dissent unleashes the regenerative capacities that allow societies to prosper and is understood as a duty of citizenship [40]. Incivility can be understood as a form of protest [41], as an instrument to show the sense of injustice and promote social change. In this way, incivility would be the consequence of a rational deliberative process and, therefore, would reflect the human capacity to make decisions freely, even when they are morally wrong, to the detriment of social improvement. Finally, the moral intensity [42] of uncivil behaviors may play a significant role in their relationship with humanity. Uncivil behaviors could be considered low-intensity actions of the type frequently committed by most people, and these facts could increase their acceptance as “typically human” behaviors.

Exploring to what extent uncivil behaviors are conceptualized as representative of human beings is relevant because the way people frame behaviors helps us understand their affective response towards both perpetrators and victims. When harmful actions are conceptualized as part of human nature, triggering the human category might generate a more forgiving reaction to harmful actions committed against the ingroup. Specifically, it is expected that there will be a greater understanding of the harmful behavior and a reduced tendency to blame the group as a whole. Wohl and Branscombe [43] found that participants display more forgiveness towards an outgroup that committed atrocities towards their ingroup in the past when these atrocities are presented as an illustration of “what humans do to each other” instead of what a specific outgroup did to the ingroup. Moreover, perpetrator groups are exculpated to a higher degree because their actions are examples of how human beings typically behave toward one another. Framed in this way, harmful actions become easier to justify and less worthy of guilt. Furthermore, using a negative view of humanity to explain or describe harmful behavior legitimizes violence [44].

To sum up, from a wide variety of perspectives, one could expect that uncivil behaviors are conceptualized as inherently human or, on the contrary, as inhuman behaviors. The present research sought to answer this question and examine the automatic associations that people establish between humanity and both civic and uncivil behaviors to explore whether the conceptualization of the behaviors follows a similar or different pattern of association than the perception of the perpetrators.

To represent humanity, several stimuli can be used. For example, previous research has shown an automatic association between secondary emotions and human parts and primary emotions and animal parts [45]. Viki and colleagues [46] used animal-related and human-related words to capture humanity. Taking into account these studies, different stimuli representing humanity were selected for the present research. In the first study, a set of animal and human pictures was presented. The second and the third study used animal-related vs. human-related words (see Table 1). The appendix shows examples of the stimuli presented.

All the studies are designed to test whether a greater association will be established between civic behaviors and humanity (represented by human pictures and human-related words) than between uncivil behaviors and humanity. In recent decades, implicit measurement techniques have undergone considerable development, and the implicit association test (IAT; [47]) is the most widely used due to its robustness, reliability, and ease of administration (e.g., [48]). However, the fact that the conventional IAT is restricted to measuring relative association strengths between two concepts (for a review, see [49]) has facilitated the introduction of several variants of this instrument. The single category IAT (SC-IAT) has been developed to overcome this restriction and has shown solid support for validity and reliability as another implicit measure of social cognition [50]. Karpinski and Steinman [50] designed the SC-IAT as a two-phase variation of the IAT procedure to measure the evaluative associations with a single category or attitude object. In each phase, target words associated with the attitude object and an evaluative dimension are presented randomly. Unlike the IAT, a comparative SC-IAT can be broken down into its components, and SC-IATs may thus allow for specific conclusions to be drawn.

In study 1, human and animal pictures were associated with civic and uncivil behaviors. Study 2 replicates the procedure, but with human-related words and animal-related words instead of pictures. To verify that the effect is not due to a priming effect (both the civic and uncivil behaviors are actions performed by humans), study 3 replicates the experiment with uncivil behaviors that do not prime graphically human beings. Table 2 shows the descriptive statistics for participants’ sociodemographic characteristics included in each study after data reduction.

## 2. Study 1

This research aims to explore the spontaneous associations between civic behavior, incivilities and humanity. In the first study, we selected human and animal pictures and pictures representing civic and uncivil behaviors. With the SC-IAT procedure, we expect to test whether civic behaviors and human pictures had a stronger automatic association than uncivil behaviors and human pictures.

### 2.1. Methods

#### 2.1.1. Participants

Participants were a total of 64 undergraduate students, all residents of Spain. Participants’ ages ranged from 18 to 48 (M age = 19.51; SD age = 4.01); 52 were female and 12 were male. Participants were students recruited on a University Campus. Specifically, the researchers attended first-year classes in psychology and social work and asked their students for their voluntary participation in exchange for partial course credit. No exclusion criteria were applied. All the participants gave informed consent and reported that they understood the experimental procedure.

#### 2.1.2. Materials

Civic and uncivil pictures. All the behaviors were extracted from the pretest study of civic and uncivil behaviors by Rodríguez-Gómez and colleagues [51] in which a total of 120 behaviors were evaluated in the civility dimension. An illustrator transformed all the written behaviors into pictures. A group of 85 collaborators made a description of each of the vignettes. A group of inter-judges evaluated whether the description of the vignettes of each subject coincided with the conduct label of the study by Rodríguez-Gómez and colleagues [51]. When the vignettes had more than 75% correspondence with the original behaviors, they were considered correctly represented visually. Seven uncivil pictures with the highest percentage of success were chosen. The same was done for the civic pictures.

The seven target pictures about civic behaviors employed in the civic SC-IAT were giving your seat up to an older person (M_civility_ = 5.00, SD = 0.00), helping push a broken-down car (M = 4.84, SD = 0.37), recycling glass (M = 4.48, SD = 0.95), respecting parking places reserved for people with disabilities (M = 4.07, SD = 1.58), turning off your mobile phone in the cinema (M = 3.89, SD = 1.29), picking up dog droppings (M = 3.88, SD = 1.69), and taking out the garbage at the set time (M = 3.72, SD = 0.92). The seven target pictures about uncivil behaviors used for the uncivil SC-IAT were throwing papers and trash on the street (M = 1.25, SD = 0.71), ruining the street furniture (M = 1.32, SD = 0.88), not picking up dog droppings (M = 1.39, SD = 0.85), not giving your seat up to an older person (M = 1.43, SD = 0.92), emptying your car ashtray onto the street (M = 1.69, SD = 1.32), leaving garbage outside the container (M = 1.81, SD = 1.29), and parking in a parking space reserved for people with disabilities (M = 1.85, SD = 1.09) (see Appendix A). Each target picture was 500 pixels in width and 375 pixels in height.

Before the SC-IAT procedure, a familiarization task with the vignettes was carried out. The participants were presented with a booklet with the images, and they had to indicate using 3 response alternatives what behavior was represented in each of the images. The aim was to ensure that participants identified the behavior represented in each picture.

Human and animal pictures. Forty-two pictures were used (see Appendix B). All images were extracted from the study by Bates and colleagues [52], which includes an extensive collection of drawings and a set of data related to each one. For this study, 21 images corresponding to animals and 21 images of human beings were selected that represented different professions, were performing some action or simply represented an age group (e.g., a baby, a girl). This selection was made looking for there were no significant differences in the proportion of valid responses to describe the image (M = 0.88; SD = 0.20 and M = 0.81; SD = 0.19, respectively for the images of animals and humans; *t*_(40)_ = 1.12; *p* = 0.271). Likewise, the selection was made taking into account the reaction times that the images needed to generate a valid response (M = 1003.62; SD = 198.85 and M = 1002.43; SD = 231.12, respectively for the images of animals and humans *t*_(40)_ = 1.485; *p* = 0.145).

#### 2.1.3. Procedure

SC-IAT measure of civic and uncivil associations. Each participant had to perform two SC-IAT procedures, one for the target category of civic behavior and another for uncivil behavior. The order of presentation was balanced between participants. The civic SC-IAT had civic behavior as the target category and consisted of two stages, whose order of appearance was balanced across participants. Each stage consisted of 24 practice trials, which familiarized the participant with the procedure, but was not taken into account in the analysis, immediately followed by 72 test trials (presented in three blocks of 24 trials). In the first stage (civic—human pictures), participants had to press the *p* key the moment they saw civic behaviors and human pictures on the screen, and they had to press the *q* key when they observed animal pictures. The dependent variable was reaction time for each stimulus, measured in milliseconds. To correct a response bias, civic behaviors, human pictures, and animal pictures were not presented with identical frequency. Specifically, a 7:7:10 ratio was selected. Hence 58% of correct responses were on the *p* key, and 42% of correct responses were on the *q* key [48].

In the second stage (civic—animal pictures), human pictures were categorized on the *p* key, and civic behaviors and animal pictures were categorized on the *q* key. Civic behaviors, human pictures, and animal pictures were presented in a 7:10:7 ratio so that 42% of correct responses were on the *p* key and 58% of correct responses were on the *q* key. Within each category, pictures were selected randomly without replacement.

A set of instructions concerning the categorization task and the appropriate key responses preceded each stage. After this, each target picture appeared in the center of the screen (500 × 375). Category reminder labels were positioned at the top of the screen. The target item remained on the screen until the participants responded or for 1500 ms. After each response, feedback about the accuracy of each response appeared. Correct responses were followed by a green O in the center of the screen for 150 ms, whereas incorrect responses were followed by a red X in the center of the screen for 150 ms.

For the uncivil SC-IAT, the procedure was repeated with the target category uncivil behavior and using the same human and animal pictures as in the first condition. The design of the experiment was within-subject; that is, all the participants performed both the civic SC-IAT and the uncivil SC-IAT. The presentation of the civic and uncivil SC-IAT was balanced, meaning that approximately half of the participants completed the civic SC-IAT first, and the other half performed the uncivil SC-IAT first. All measures, exclusions, and manipulations in the study are reported in this manuscript.

### 2.2. Results

Compared with the IAT, error rates are usually significantly higher in the SC-IAT. While the response window in the SC-IAT procedure facilitates quick responding, this is likely to be accompanied by increased error rates. For this reason, the same data reduction procedure of Karpinsky and Steinman [50] was used. Participants with an error rate greater than 20% on the civic or uncivil SC-IAT were excluded from the analysis, resulting in eliminating five participants (average error rates: civic SC-IAT = 7.24%; uncivil SC-IAT = 6.23%). Responses under 350 ms were eliminated, nonresponses were eliminated, and erroneous responses were replaced with the block mean plus an error penalty of 400 ms.

Following the procedure described by Karpinsky and Steinman [50], a scoring algorithm was modeled on the *D*-score algorithm (proposed by Greenwald et al. [47]). The average response times of block 2 (civic—human pictures) were subtracted from the average response times of block 1 (civic—animal pictures). This quantity was divided by the standard deviation of all correct response times within blocks 1 and 2. Thus, positive SC-IAT D scores indicate stronger associations for human pictures than for animal pictures for civic and uncivil behaviors. As a result, two SC-IAT *D-*scores were calculated, one for civic pictures and one for uncivil pictures. For each of the D scores, a *t-*test comparison is conducted to test whether the score differs from 0.

For the civic condition, the SC-IAT revealed that participants had more human associations with civic behaviors than animal associations (*d* = 0.257; *t*_(58)_ = 10.05; *p* < 0.001; CI for difference = 0.206, 0.308). The same results were obtained in the uncivil condition: participants had more human associations with uncivil behaviors than animal associations (*d* = 0.251; *t*_(58)_ = 11.03; *p* < 0.001; CI for difference = 0.205, 0.296) (see Figure 1).

### 2.3. Discussion

The first study showed that the automatic association between civic behaviors and human pictures was faster than between civic behaviors and animal pictures. The same pattern was obtained for uncivil behaviors. This result does not confirm the idea extracted from Haslam’s model of dehumanization that considers lack of civility as a characteristic of dehumanization. When participants think in terms of behaviors, both civic and uncivil behaviors were strongly associated with humanness, compared with animal concepts.

## 3. Study 2

The second study aims to replicate the results with different human and animal stimuli. Specifically, we selected human and animal words and the same pictures representing civic and uncivil behaviors as those used in study 1. With the SC-IAT procedure, we expect to test whether civic behaviors and human words have a stronger automatic association than uncivil behaviors and human words.

### 3.1. Methods

#### 3.1.1. Participants

A total of 64 Spanish undergraduate students (12 male, 52 female) participated voluntarily in this study. All received course credit for their participation in the research. Participants’ ages ranged from 18 to 40 (M age = 19.42; SD age = 3.04). Participants were students recruited on a University Campus. Specifically, the researchers attended first-year classes in psychology, social work, nursing, and labor relations and asked their students for their voluntary participation in exchange for partial course credit. No exclusion criteria were applied. All the participants gave informed consent and reported that they understood the experimental procedure.

#### 3.1.2. Materials

Civic and uncivil pictures. The same civic and uncivil pictures were used as in study 1.

Human-related and animal-related words. 21 human-related words (e.g., wig, flag, symbol, parliament, gang) and 21 animal-related words (e.g., cage, beast, sting, bug, hoof) were used (see Appendix C). All the words were selected from a pilot study carried out with 54 people (M_age_ = 24.48; SD = 6.86) divided into two samples that assessed the extent to which 90 words were associated with something animal (1) or something human (7) and in which they were negative (1) or positive (7). The t-test comparison of the 42 words showed that there were significant differences in humanity (*t*_(40)_ = 25.98; *p* <.001), but not in valence (*t*_(40)_ = 0.17; *p* = 0.627) between the two groups of words. Each word was presented in 44pt Courier New font.

#### 3.1.3. Procedure

SC-IAT measure of civic and uncivil associations. The procedure was identical to that used in study 1, except for the target words. Animal-related words and human-related words replaced the labels animal pictures and human pictures, respectively, with 21 target words used for each. Each word appeared on the screen in 44pt Courier New font. All measures, exclusions, and manipulations in the study are reported in this manuscript.

### 3.2. Results

Participants with an error rate greater than 20% on the civic or uncivil SC-IAT were excluded from the analysis, resulting in the elimination of 13 participants (average error rates: civic SC-IAT = 9.40%; uncivil SC-IAT = 8.62%).

SC-IAT scores were computed by using the scoring algorithms described for study 1. Thus, SC-IAT *D* scores indicate stronger associations for human-related words than for animal-related words for civic and uncivil behaviors. The SC-IAT revealed that participants had more human associations with civic behaviors than animal associations (*d* = 0.109; *t*_(50)_ = 3.27; *p* = 0.002; CI for difference = 0.042, 0.176). The same results were obtained for uncivil behaviors: participants had more human associations with uncivil behaviors than animal associations (*d* = 0.163; *t*_(50)_ = 6.45; *p* < 0.001; CI for difference = 0.112, 0.213) (see Figure 2).

### 3.3. Discussion

The results obtained in this study confirmed those obtained in study 1. Specifically, the automatic association between civic behaviors and human pictures was faster than between civic behaviors and animal pictures and similar to the pattern obtained for uncivil behaviors. Again, participants showed an automatic association between uncivil behaviors and human words, which did not support the idea extracted from Haslam’s model of dehumanization that considers lack of civility as a characteristic of dehumanization.

## 4. Study 3

Studies 1 and 2 showed a very similar pattern of automatic association between civic and uncivil behaviors and human and animal stimuli. However, the effect may be due to a priming effect. One possible alternative explanation of our results is that the participants simply indicated that both the civic and uncivil behaviors are actions performed by humans. That is, in an associative task, such as the SC-IAT, and uncivil behavior could be related to human words or pictures simply because it is an act performed by people. Both civic and uncivil stimuli primed humans. In the pictures, human beings appeared doing civic/uncivil behaviors. The purpose of the third study was to replicate the previous results, but with civic/uncivil behaviors that do not prime graphically human beings.

### 4.1. Methods

#### 4.1.1. Participants

A total of 67 Spanish undergraduate students (12 male, 55 female) participated voluntarily in this study. All received course credit for their participation in the research. Participants’ ages ranged from 18 to 26 (M age = 20.09; SD age = 2.11). Participants were students recruited on a University Campus. Specifically, the researchers attended first-year classes in psychology and asked their students for their voluntary participation in exchange for partial course credit. No exclusion criteria were applied. All the participants gave informed consent and reported that they understood the experimental procedure.

#### 4.1.2. Materials

Civic and uncivil pictures. The civic and uncivil pictures used in Studies 1 and 2 were modified to eliminate persons from the picture. The same behavior was presented without including in the picture any person.

The civic behaviors of giving your seat up to an older person and helping push a broken-down car from study 1 could only be represented by a human agent, so these civic behaviors were replaced by flushing the toilet after using a public washroom (M = 4.44, SD = 0.85) and throwing your cigarette butts in the rubbish (M = 4.07, SD = 1.27). In uncivil behaviors, the same occurs with not giving your seat up to an older person, parking in a parking space reserved for people with disabilities and emptying your car ashtray onto the street, which are replaced by painting graffiti on street furniture (M = 1.54, SD = 1.27), not flushing the toilet in a public washroom (M = 1.46, SD = 0.82), and washing your car on the street (M = 2.92, SD = 0.49) (see Appendix D).

Human-related and animal-related words. The same words as those in study 2 were used.

#### 4.1.3. Procedure

SC-IAT measure of civic and uncivil associations. The procedure was identical to that used in study 2, except for the civic and uncivil pictures. Each word appeared on the screen in 44pt Courier New font. All measures, exclusions, and manipulations in the study are reported in this manuscript.

### 4.2. Results

Participants with an error rate greater than 20% on the civic or uncivil SC-IAT were excluded from the analysis. In this study, no participant was eliminated (average error rates: civic SC-IAT= 7.11%; uncivil SC-IAT= 7.01%).

SC-IAT scores were computed by using the scoring algorithms described for Studies 1 and 2. Thus, SC-IAT *D* scores indicate stronger associations for human-related words than for animal-related words for civic and uncivil behaviors. For civic behaviors, the SC-IAT revealed that participants had more human associations with civic behaviors than animal associations (*d* = 0.440; *t*_(66)_ = 4.19; *p* < 0.001; CI for difference = 0.230, 0.651). The same results were obtained in study 2: participants had more human associations with uncivil behaviors than animal associations (*d* = 0.485; *t*_(66)_ = 4.46; *p* < 0.001; CI for difference = 0.268, 0.702) (see Figure 3).

### 4.3. Discussion

The purpose of this study was to replicate the previous results, but with civic/uncivil behaviors that do not prime graphically human beings. Results confirmed the strong association between both civic and uncivil behaviors and human words. This study disconfirms the idea that results could be explained by a mere priming effect.

## 5. General Discussion

While previous literature has focused on the human perceptions of perpetrators of harmful or immoral acts, little is known about the links between human perception and harmful behaviors. This research aimed to explore the automatic associations between humanity and civic and uncivil behaviors. The results of the three studies consistently supported an association between humanity and both types of behaviors across the different stimuli used to represent the concept of humanity.

Specifically, in study 1, in which pictures of humans and animals were used, the results indicated stronger associations between human pictures and civic and uncivil behaviors than animal pictures. In study 2, animal-related words and human-related words replaced the animal and human pictures, and the results were exactly the same: participants had stronger associations between human-related words and both civic and uncivil behaviors than with animal-related words. In study 3, the same pattern of association was found with civic and uncivil behaviors represented without, including human beings.

Haslam’s dual model of dehumanization [21] proposes that civic behaviors are considered uniquely human (UH) traits, whereas incivilities are the expression of a lack of these traits and hence should be associated with animalization. Taken as a whole, our results confirm that the notion of humanity is associated not only with civic behaviors but also with uncivil behaviors. These results could be interpreted in line with previous literature suggesting that negative actions are linked with agency and consequently with humanness [25]. Agency is a core element of attributions of humanness [53] and could be associated with both positive and negative behaviors [25].

Previous research has shown that the association between humanity and negative behaviors occurs and that this association has important consequences for the perception of such behaviors. For example, Wohl and Branscombe [43] found that representatives of a victimized group show greater willingness to forgive past harm when their oppression is categorized as a prototype of “what humans do to each other” (i.e., when the human category becomes salient). In the same line, Morton and Postmes [44] proposed that considering one’s flaws and failings as “uniquely human” may be stimulated by the wish of reducing individual responsibility for these failings and thus diminish their effect on self-regard.

Furthermore, there are several characteristics of uncivil behaviors that could facilitate their link with humanity, compared with other harmful behaviors. One is their degree of moral intensity [42]. In this sense, uncivil behaviors could be considered actions resulting in low-intensity damage, which is to say, actions of low moral intensity. Moral intensity has a direct impact on ethical responses [42]. On one hand, it affects the intention of deciding to behave in an ethical or unethical way and also the decision to engage in the action. A behavior with very high moral intensity creates pressure to make a more ethical decision, while in everyday situations with low moral intensity, people may feel little threat in behaving immorally [54]. On the other hand, moral intensity affects the awareness that an action presents an ethical dilemma and whether this action is morally right or wrong. With low moral intensity, the moral standards are not so clear. In this case, people may violate moral norms because they have convinced themselves that their behavior is not so immoral or are even not aware that it violates moral principles. The same may happen with the perception of behavior. If one considers uncivil behaviors as low-intensity deviant behaviors with ambiguous intent to violate the social norms for mutual respect [18,55], it may be that the uncivil behaviors are not clearly seen as an extreme example of a violation of the rule or moral principle (immoral behaviors), but rather as “typically human” behaviors. In this sense, one could expect to find an association between incivilities and human terms. In this line, more studies are needed to explore the various components of moral intensity [42] and their effect on the perception of uncivil behavior. It would be interesting to study the role of the magnitude of consequences, the degree of social agreement that an act is bad (social consensus), the probability that the action is carried out and that it causes the damage (probability of effect), the temporal immediacy of the consequences, the proximity between the actor and the victim, and the extent to which the act creates a general good for a greater number of people (concentration of effect). Following this last concept, it would be interesting to study the extent to which the conception of uncivil behavior as an instrument for improving the common good of society, that is, as a behavior with a high effect concentration, affects its association with the dimension of humanity.

Our results consider whether judging the behavior and judging the person who performs that behavior is two different processes. In line with what Khamitov and colleagues [28] suggested, it is possible that if the focus of attention is placed on the person, who performs the behavior instead of on the action, the agent could be negatively evaluated and considered less human. Nevertheless, when people judge the behaviors, the automatic association between uncivil behavior and humanity remains. Following this reasoning, incivilities can be considered typically human and, at the same time, the person who commits these acts can be perceived as not fully human. Undoubtedly, more research is needed to clarify alternative explanations. It is important to remark that the present study does not explore the implicit association between uncivil agents and humanity, and conclusions should be limited to this specific aspect.

This study has both theoretical and practical implications. On one hand, our results have implications for social psychology, specifically for dehumanization theory [21,25]. Research examining the association between incivility and humanness is scarce; to our knowledge, prior research has not directly assessed the empirical association between humanness and both civic and uncivil behaviors, testing the theoretical argument that uncivil behaviors are less associated with the concept of humanness than civic behaviors. This study may fill this theoretical gap in the literature on dehumanization.

In addition, the study of automatic associations between humanness and incivilities could help to explore the reasons that lead people to justify and exculpate this type of harmful behavior. Several studies support adding dehumanization to models of justice [29,56,57,58]. Dehumanization accounts for how perceptions of the harmfulness of deviant behavior are translated into a desire for severe forms of punishment; that is to say, the perceived inhumanity of the perpetrators is found to be an important determinant in the judgment of blame and punishment. However, when the opposite occurs, when human nature is portrayed as fundamentally malevolent, that is, when the human category becomes salient and humanity is associated with deviant behavior, negative forms of action may be normalized and thus legitimized [44]. In this way, associating an uncivil behavior with humanness, that is, understanding this behavior to be typical of human beings, may affect its acceptability and thus lead to decreased social control and punishment.

This study opens up new opportunities for better understanding the mechanism underlying the social perception of uncivil behaviors. Interestingly, uncivil behaviors are sometimes conceptualized as a way to confront the status quo and the power of privileged members of society. This association of uncivil behavior with the typically human may inhibit the self-regulation processes associated with the demands of social life and respect for the environment, and this tendency to indiscriminately humanize both the civic and the uncivil may contribute to uncivil actions being used as a “legitimately human” way of responding to injustice, economic inequality, or power differences. The destruction of street furniture as a form of protest (for example, in Hamburg on the occasion of the G20 summit or in Barcelona due to secessionist demands) is a good example of this.

Finally, the findings of this study could help planners and policymakers with the design of campaigns to prevent uncivil behaviors, considering the way people conceptualize and even justify this type of behavior. Actions intended to increase respect for the rules of coexistence and for the environment cannot be limited to explicit messages or campaigns that demand greater awareness of citizens. These campaigns are supported by explicit studies on public opinion that show strong support for the negative assessment of uncivil behaviors and an unpleasant emotional reaction of discomfort and stress (e.g., [38,59]). However, the results of this research show that there is an implicit and automatic tendency to humanize both civic and uncivil behaviors, that is, to perceive them indiscriminately as typically human actions. Our study supports the need to employ more covert persuasion strategies (e.g., [60,61]) that act on the implicit and unconscious attitudes about transgressions to society and the environment.

The present work has several limitations that should be considered and that offer interesting opportunities for future research. First, a single-measure instrument has been used. It would be interesting to check whether the same results are obtained with different implicit measures (e.g., affect misattribution procedure, go/no-go association task, or identification extrinsic affective Simon task) and compare them with deliberative or explicit responses. Second, a limited range of civic and uncivil behaviors was used. In addition, not all the uncivil behaviors used in the studies affect others to the same degree. That is, while all are deemed negative behaviors in that they infringe social norms, they differ in the extent to which they cause damage to others. Rodríguez-Gómez and colleagues [51] provide an extensive database of 120 civic and uncivil behaviors evaluated in various dimensions relevant to civility and humanness. From this database, it is possible to extract different behaviors according to research interests. Since the social repercussion of the behaviors is among the dimensions evaluated in this database, new studies can select behaviors representative of this dimension to control the effect of harm on others. Concerning the sample used, we limited our study to university students from a particular region. More research is required to confirm that the results can be replicated in an older population, as well as to determine whether national or cultural differences could influence the results obtained. While studies confirm that there is no gender effect in anticipation of incivilities (women are no more likely than men to anticipate uncivil behaviors) (e.g., [62,63]), it also has been shown that women’s perceptions of crime prevalence are significantly higher than men’s (e.g., [64,65]) and that women are more critical and sensitive than men concerning the quality of the public space [66]. In this vein, the fact that a large percentage of the sample were women may influence the perception of the humanness of the behaviors studied. Future studies should replicate the results with a balanced sample in the number of men and women. Another limitation of the studies is the low number of participants. The association between civic and uncivil behaviors and humanity may seem evident since both types of behaviors are committed by human beings. However, previous research with perpetrators of these actions has shown that this is not the case. In the study of agent perception, it has been found that dehumanization occurs even when the agents are humans committing human acts.

Despite these limitations, the current research provides empirical evidence to better understand the lay conceptualizations of incivilities and opens up new possibilities for exploring the associations between uncivil behavior and humanness. Future research could include the severity of the consequences of uncivil behavior to verify whether this influences the perception of the humanity of incivilities. Further research may also study the role of perception of damage on the association between humanity and incivilities. Additionally, future studies should explore whether other negative behaviors are associated with humanity or whether they are more related to animal terms, even if they are committed by human beings. Importantly, the study of social perception of civility should be circumscribed to the specific cultural context where the study is carried out. Relevance, frequency, and justification of each behavior could vary considerably across cultures. Replication of the results with samples from other countries should help us to better understand the automatic associations between humanness and uncivil behaviors. Finally, the degree to which participants in the studies have suffered or perpetrated uncivil behaviors (perceived frequency) is another relevant issue that should be included in future studies. In fact, psychological distance (temporal, spatial, likelihood, social) [67] can modulate the perception that a moral violation is more or less benign [68] and therefore, it may influence our conception of how typically human it is.

## 6. Conclusions

In conclusion, the present study extends previous research on dehumanization theory and has direct implications for the conceptualization of humanness for finding a consistent association between this dimension and both civic and uncivil behaviors. Our results show that there is an automatic association between uncivil behaviors and humanness. Implications of these findings could be related to accepting these behaviors since harmful actions become easier to justify and less worthy of guilt when they are framed as typically human actions. Future research will help to offer a broader understanding of the underlying processes and relevant factors that modulate and clarify these automatic associations.

## Figures and Tables

**Figure 1 ijerph-18-04353-f001:**
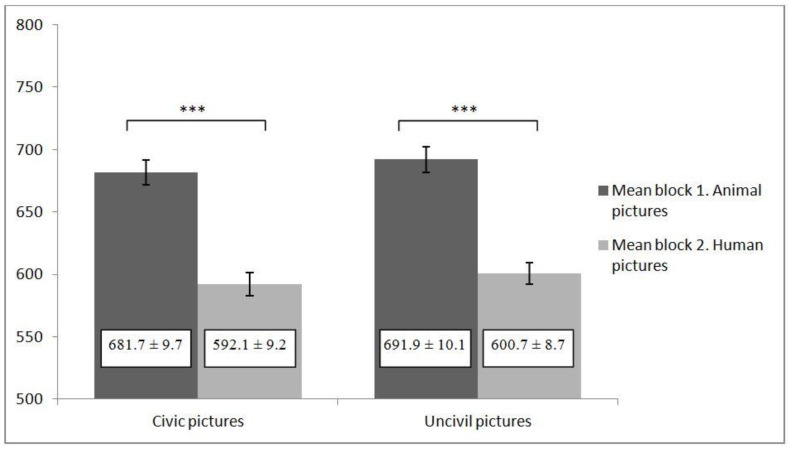
Means ± standard error for reaction times (in milliseconds) of block 1 and 2 in study 1. *** indicated *p* < 0.001.

**Figure 2 ijerph-18-04353-f002:**
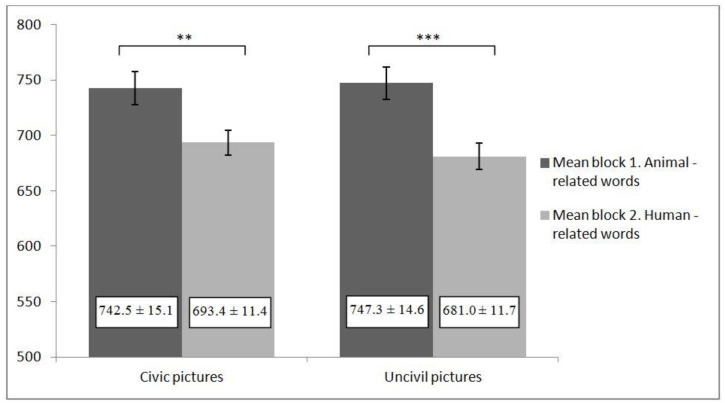
Means ± standard error for reaction times (in milliseconds) of block 1 and 2 in study 2. *** indicated *p* < 0.001, ** indicated *p* < 0.01.

**Figure 3 ijerph-18-04353-f003:**
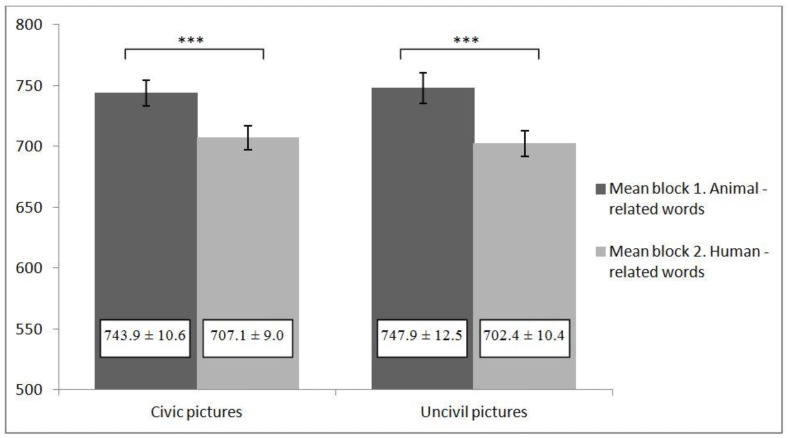
Means ± standard error for reaction times (in milliseconds) of blocks 1 and 2 in the study. *** indicated *p* < 0.001.

**Table 1 ijerph-18-04353-t001:** Overview of the studies.

Study	Target	Associative Words
Study 1	Civic and uncivil behaviors (human agent)	Animal vs. human pictures
Study 2	Civic and uncivil behaviors (human agent)	Animal vs. human-related words
Study 3	Civic and uncivil behaviors (not human agent)	Animal vs. human-related words

**Table 2 ijerph-18-04353-t002:** Descriptive statistics for sociodemographic characteristics of each study after data reduction.

	Study 1 (*N* = 59)			Study 2 (*N* = 51)			Study 3 (*N* = 67)		
	*n* (%)	M (SD)	Range	*n* (%)	M (SD)	Range	*n* (%)	M (SD)	Range
Age		19.51 (4.15)	18–48		19.42 (3.04)	17–40		20.09 (2.11)	18–26
Gender									
Female	48 (81.4)			49 (86.3)			55 (82.1)		
Male	11 (18.6)			7 (13.7)			12 (17.9)		
Degree									
Psychology	32 (54.2)			24 (47.1)			67 (100)		
Social work	37 (45.8)			20 (39.2)					
Nursing				5 (9.8)					
Labor relations				2 (3.9)					

In study 1, five participants were eliminated. In study 2, 13 participants were eliminated. In study 3, no participant was eliminated.

## Data Availability

The datasets generated for this study are available at 10.17632/6tb9zv67kg.1.

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
