# Peer review of "Humanness Is Not Always Positive: Automatic Associations between Incivilities and Human Symbols"

_ijerph, 2021, doi:10.3390/ijerph18084353_

Round 1

Reviewer 1 Report

The paper aims at a study of implicit associations between civicity and human symbols, measured by means of SC-IAT tests using graphical and verbal stimuli. The main idea of the paper is clear and properly stated. Literature review is appropriate, although could have been extended.

While the idea of the experimental design is clear, my concern is whether it really shows what the authors imply it should. The SC-IAT technique as developed by Karpinski and Steinman (2006) is based on categories that can be associated with both alternatives (Good words-self words vs Bad words, and the like). By contrast, the alternatives used in the present paper are human or animal, while the categories of civicity are human par excellence: animals don’t park cars, give up seats in public transport or flush the toilet as a rule. Hence it is not at all surprising that subjects associate both civic and uncivic behavior with humans rather than animals. The authors claim that the results obtained reject the `animalization’ part of the dual dehumanization theory by Nick Haslam. To me, this conclusion is based on misinterpretation of Haslam: the meaning of `animalization’ is not that denumanized people are `like animals’, it is that they are like animals in particular (usually negative) connotation or context: “dirty as a pig”, “stubborn as a donkey”, “walks like a monkey” etc. Animal pictures and words used in the study do not convey such connotations, hence they should not be claimed to test Haslam’s theory either.

A question of real interest that might have been asked in this context is whether uncivic behaviour is associated with particular subgroups or ethnicities which behave themselves `like animals’: are not clean, polite, respectful to social norms etc. This question may be tackled with SC-IAT like techniques, e.g. showing pictures featuring people of different ethnicities (in different treatments), committing an uncivic act, and then comparing them to specific animals. If associations with animals will be stronger for some ethnicities than for others, that would be evidence of discrimination, and dehumanization may be treated as one of the mediators of the connection between discriminating attitudes and attribution of uncivic behaviour to particular ethnicity. It is not clear ex ante whether or not this connection will be confirmed – but in either case, it would be another study.

Design description is sometimes unclear: the main experiment of Study 1 should apparently read as two  replications of 72 trials each (civic-human vs. animal and civic-animal vs human), but it reads otherwise - as two blocks of 24 trials each, leaving it unclear where is the third block.

From the text, it is not clear on what grounds “Responses under 350 ms were eliminated, 280 nonresponses were eliminated, and erroneous responses were replaced with the block 281 mean plus an error penalty of 400 ms.” (lines 280-282).

Figure 1 (and subsequent figures), along with means, feature whiskers which are usually associated with standard errors – yet under these interpretation, s.e. are too large for significant treatment effects. Annotation of these figures should be complete and self-sufficient.

Some proofreading would be helpful. For instance, first introduction of abbreviation Single Category IAT (CS-IAT, line 172) appears before abbreviation IAT (line 174).

Reviewer 2 Report

The manuscript ijerph-1148648 entitled Humanness is not always positive: Automatic associations between incivilities and human symbols uses 3 studies to examine the automatic associations that people establish between humanness and both civic and uncivil behaviors.

Although the authors describe the 3 studies subsequently, a common methods section is missing. This would help to explain the study design in full detail (recruitment, selection criteria, response rate….) and the chronological sequence of the studies. Also a table summarizing socio-demographic characteristics of study subjects is missing.

Practical and theoretical implications of the overall findings are lacking, especially clarity is needed on how these Spanish data can be reflected in international settings.

It seems unusual that the main text is ranged right.

Please do not use the keywords from the title. Also, I wonder why the keywords are numbered.

Please be consistent in writing style such as in e.g. Wig, flag, Symbol.

Shortcomings related to study participants` gender distribution and very small number of participants should be discussed.

Reviewer 3 Report

The article presents research on the relationship between humaneness and civic and uncivil behaviour. The results obtained in three studies indicate that, although the state of the art in academia links uncivil behaviours with dehumanisation, research participants associate both types of behaviour with human action.

The introduction provides sufficient background on the topic and previews major points. Both research design and analysis are adequate.

The discussion and conclusion sections contribute greatly to the explanation of the results.

I recommend extending the section on conclusions.

Reviewer 4 Report

I think this is a highly important and interesting paper. The chosen methodology allowed analysing the selected research object and presenting conclusions.

The article is prepared properly and clearly; therefore, I note just a few details that I think need to be clarified:

In the introduction, I would recommend reasoning the ideas of the article more clearly with the general ideas and philosophy of the journal. When reading, it is not clear why and how this issue relates to the general ideas of the journal.

Three studies are presented, but the method of selection of participants thereof is not clear enough. It is not clear if there are the same participants (from the context, I guess they are not). It is necessary to give a more detailed explanation of how the study participants were selected, why they were selected, what criteria determined their selection, and so on. How did the researchers consider the overall analysis of the results if different people were involved in all stages? Could this have any influence on the results?

When the limitations of the study are presented in the discussion part, it is slightly confusing to read, so I would recommend distinguishing the limitations of the study and the prospects for further research.

The conclusions are presented in a very general way, focusing on the prospects for further research. I would recommend supplementing them and providing the basic insights together with the main insights from the study results.

Round 2

Reviewer 2 Report

I thank the authors for modifying their manuscript to adress issues raised in the first review round.

However, unfortunately, most of the newly added content contains further shortcomings such as:

+ what is meant by "a University Campus" in "Participants were recruited in a University Campus."?

+ If there would be no exclusion criteria ("No exclusion criteria were applied.), how could be ensured to target the correct study population?

+ Another limitation of the studies is the number of participants. : high or low number....?

The figures are not very meaningful and could be rescaled or omitted, as the resulats are quoted in the text.

Ideas are lacking how to combat the limitations that are added.

The paragraph on theoretical and practical implications has potential, but references and concrete strategies are lacking.

Please correct: Table 2. ...after the reduction data.

..elimination of 0 participants.

...one for the target category of civic behavior and other for uncivil
behavior.

Among the mentioned  ones, there are others so extensive editing of English language and style is required, also check the tenses!
